# A scoping review of the electronic collection and capture of patient reported outcome measures for children and young people in the hospital setting

**Anne Alarilla**[1,2]*, **Neil J. Sebire**[1,3], **Josh Keith**[2], **Mario Cortina-Borja**[1], **Jo Wray**[4,5‡], **Gwyneth Davies**[1,6‡]

**1** UCL Great Ormond Street Institute of Child Health, London, United Kingdom, **2** The Health Foundation, London, United Kingdom, **3** NIHR Great Ormond Street Hospital for Children NHS Foundation Trust Biomedical Research Centre, London, United Kingdom, **4** Centre for Outcomes and Experience Research in Children's Health, Illness and Disability (ORCHID), Great Ormond Street Hospital for Children NHS Foundation Trust, London, United Kingdom, **5** Institute of Cardiovascular Science, University College London, London, United Kingdom, **6** Heart and Lung Directorate, Great Ormond Street Hospital for Children NHS Foundation Trust, London, United Kingdom

‡ These authors are joint senior authors on this work.
* anne.alarilla.16@ucl.ac.uk

## Abstract

Patient reported outcome measures (PROMs) capture patients' views of their health status and the use of PROMs as part of standard care of children and young people has the potential to improve communication between patients/carers and clinicians and the quality of care. Electronic systems for the collection of or access to PROMs and integrating PROMs into electronic health records facilitates their implementation in routine care and could help maximise their value. Yet little is known about the technical aspects of implementation including the electronic systems available for collection and capture and how this may influence the value of PROMs in routine care which this scoping review aims to explore. The Joanna Briggs Institute review process was used. Seven databases were searched (Emcare, Embase MEDLINE, APA PsychInfo, Scopus and Web of Science), initially in February 2021 and updated in April 2023. Only studies that mentioned the use of electronic systems for the collection, storage and/or access of PROMs as part of standard care of children and young people in secondary (or tertiary) care settings were included. Data were analysed using frequency counts and thematically mapped using basic content analysis in relation to the research questions. From the 372 studies that were eligible for full text review, 85 studies met the inclusion criteria. The findings show that there is great variability in the electronic platforms used in the collection, storage and access of PROMs resulting in different configurations and fragmented approaches to implementation. There appears to be a lack of consideration on the technical aspects of the implementation such as the accessibility, useability and interoperability of the data collected. Electronic platforms for the collection and capture of PROMs in routine care of CYP is popular, yet, further understanding of the technical considerations in the use of

**Data Availability Statement:** The data extracted from the studies and analysed within this review can be found in Table 2 and S4 File.

**Funding:** AA is funded on a Clinical Informatics Research Programme PhD studentship by Great Ormond Street Hospital Children's Charity (Award VS0618) (https://www.gosh.org/). GD is supported by a personal fellowship from UK research and Innovation (UKRI FLF, MR/T041285/1) (https://www.ukri.org/). All research at Great Ormond Street Hospital NHS Foundation Trust and UCL Great Ormond Street Institute of Child Health is made possible by the NIHR Great Ormond Street Hospital Biomedical Research Centre (https://www.gosh.nhs.uk/our-research/our-research-infrastructure/nihr-great-ormond-street-hospital-brc/).

**Competing interests:** GD reports speaker honoraria from Vertex Pharmaceuticals and Chiesi Ltd, and institutional fees for clinical trial leadership and advisory board roles from Vertex Pharmaceuticals, outside the submitted work. All other authors declare no competing interests.

electronic systems for implementation is needed to maximise the potential value and support the scalability of PROMs in routine care.

## Author summary

Patient reported outcome measures (PROMs) are questions that ask about a patient's views on their health and health status, the use of PROMs as part of standard care can improve the quality of care. This scoping review maps the available evidence of the electronic systems used to collect and capture PROMs as part of the routine care of children and young people in the hospital setting. We performed a comprehensive literature search and identified 85 studies that matched our criteria for analysis. The use of electronic systems for the collection, storage and access of PROMs in routine care is popular but there are a variety of platforms available. This has led to different approaches to how the data has been collected, stored and accessed. This review highlights that it is important to consider the technical aspects of how the PROMs data collected can be accessed, shared and used across different platforms. This may help maximise the potential value of PROMs and promote further the adoption of PROMs as part of standard care in various care settings, ensuring that the patient voice is heard and represented.

## Introduction

Patient reported outcome measures (PROMs) are reports of symptoms, functioning and/or health related quality of life [1,2]. The use of PROMs in routine care can facilitate patient centred care and better communication between patients and clinicians [1,3–5]. This is particularly relevant for children and young people (CYP) who often do not feel they are asked or included in decisions about their care [6]. In the paediatric setting, PROMs can identify unmet needs, improve communication and health related quality of life, and facilitate shared decision making [7–9].

Typically, PROMs data are used at an individual level, however, the data can also be aggregated and used for other purposes such as for service planning, service improvement initiatives, research and population health monitoring [3,4,10–12]. PROMs data are therefore relevant to a range of audiences across different levels of the healthcare system from patients and their immediate care teams to researchers and policy makers [3,4,10–12]. To maximise potential across stakeholder groups, it needs to be possible to harness PROMs data as it occurs for other routinely collected data elements within electronic health records (EHRs).

The implementation of PROMs in routine care requires careful planning to ensure adoption and meaningful use [13]. Electronic systems facilitate the integration of PROMs in routine care, including within EHRs [7,13]. These electronic architectures have the potential to automate the collection process, making it easier for patients to report new or worsening symptoms [14]. They also enable real time access and presentation of results in a graphical format that is more attractive to patients and clinicians [7]. Some electronic systems also introduce the use of algorithms for clinical decision support and referrals [15,16]. Consequently, this timely access to responses can personalise care pathways, for example PROM responses can also determine whether a patient needs further contact with a clinician and whether this will be in clinic or remotely (e.g. telephone consultation) [14,16]. This may be beneficial at a system level by reducing emergency department visits and hospitalisation [14,16].

There is also an opportunity for electronic systems to support the scalability of PROMs in routine care so that the potential of PROMs data may be realised across different clinics, hospitals and nationally [12]. The integration of PROMs into EHRs adds the patient voice into the more commonly used EHR data elements (such as clinical diagnoses, laboratory results, admissions and treatments) that can be aggregated and de-identified so it is useful for secondary purposes, such as quality improvement, research including development of clinical decision support tools and clinical registries [3,4,10–12], further maximising value.

Some CYP prefer the use of electronic systems [7], whereas others prefer the use of traditional pen and paper methods [17]. Yet, the use of electronic methods for the collection of PROMs in routine care of CYP is already popular [18]. Accordingly, the use of electronic collection and capture of PROMs in routine care of CYP warrants further exploration to understand if the full potential for benefit and impact are realised. This includes a better understanding of how the electronic collection and capture of PROMs in routine care can permit different analytic approaches (individual or cohort level) that are useful within the same hospital, across different hospitals and nationally. Therefore, the aim of this scoping review was to map the available evidence on the electronic collection and capture of PROMs used as part of standard care of CYP.

We addressed the following research questions in relation to the PROMs used as part of standard care and treatment of CYP in hospitals:

- What are the electronic systems used for PROMs collection?

- What are the electronic systems used for the storage of PROMs data?

- How are PROMs data accessed electronically?

- How do the electronic collection, storage and access relate to one another?

## Methods

The focus of this scoping review was on the technical aspects of the electronic collection and capture of PROMs used in routine care of CYP. It was conducted as an extension to an overarching review which explored use of patient reported outcome measures (PROMs) and patient reported experience measures (PREMs) as part of standard care of children and young people in the hospital setting [19].

We used the Joanna Briggs Institute (JBI) methodology for scoping reviews [20]. The preferred reporting items for systematic reviews and meta-analyses extension for scoping reviews (PRISMA-ScR) [21] checklist was used for reporting.

### Eligibility criteria

The full inclusion and exclusion criteria can be found in S1 File. The eligibility criteria were established in relation to the research questions and using the Population, Concept and Context framework (PCC).

### Population

Studies were included if the data were from CYP (defined as from birth (including neonatal) to 25 years old), or their proxies (e.g., parents, carers or guardians) and/or clinicians working with CYP.

### Concept

PROMs report the patient's perception of their health status. Included studies described routine collection of PROMs data in the children's or adolescent and young adult (AYA) hospital setting with electronic collection and capture.

Electronic collection and capture of PROMs is defined as the use of electronic methods (electronic platforms, website, applications, emails) or mixed methods (pen and paper methods and electronic methods) or integration of PROMs (irrespective of method of capture) into electronic systems such as electronic health records for collection, storage, access of PROMs as part of routine care.

### Context

CYP treated by professionals based in paediatric hospital settings and CYP treated in AYA settings where data from those <25 years could be separated from those ≥25 years were included. CYP not treated in hospital-based services or other clinical setting e.g., primary care were excluded.

### Search strategy and selection criteria

The initial search was performed on 21 February 2021 and included sources published from 1 November 2008. An updated search was then performed for the period 22 February 2021 to 4 April 2023. The databases Embase, EMcare, MEDLINE, PsycINFO, CINAHL Plus (EBSCO-host), Scopus and Web of Science were searched. The final search terms including an example of a search strategy are included in S2 File.

### Study selection and charting the data

Following the searches, duplicates were removed and title and abstracts were screened by two independent reviewers against the inclusion criteria. Full text of potentially relevant sources was assessed against the inclusion criteria by two or more independent reviewers. Reasons for exclusion of sources of evidence at full text that did not meet the inclusion criteria were recorded. Disagreements between reviewers were resolved through discussion, or with an additional reviewer/s.

### Data extraction

One reviewer used a data extraction tool developed by the review team, and a minimum of 10% were reviewed by a second reviewer. The extraction form (S3 File) is based on Peters et al., (2020) [20].

### Data analysis

Data were analysed using frequency counts for different demographic characteristics of included studies and basic qualitative content analysis was used to identify themes in relation to the research questions. An inductive approach was followed as recommended by Pollock et al., (2023) [22].

## Results

The PRISMA ScR flow diagram summarizes the process of study selection (Fig 1). 85 studies met the inclusion criteria of this review [15,23–106].

Table 1 details the descriptive statistics of the included studies, S4 File reports the full characteristics of the included studies. In 76 studies (89%) PROMs were collected electronically, 5

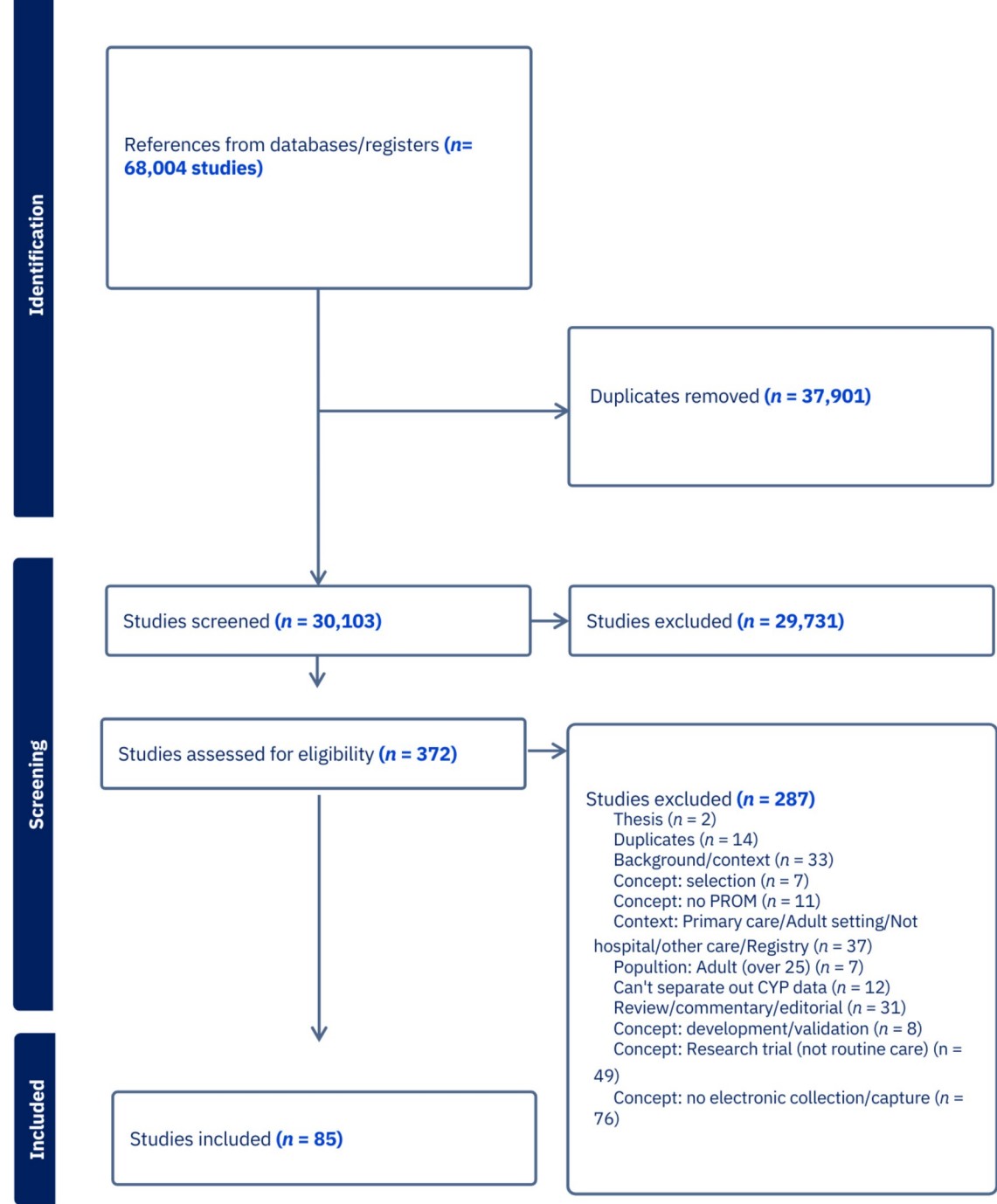

**Fig 1. PRSIMA ScR flow diagram.**

studies (5%) had mixed (pen and paper methods and electronic methods such as emails) collection and 4 studies (4%) had pen and paper collection only.

The majority of studies were from the Netherlands ($n$ = 29, 34%), followed by the United States ($n$ = 24, 28%) and Canada ($n$ = 6, 7%), the remaining countries had a count of five or less. The majority were from single centres ($n$ = 57, 67%).

**Table 1. Descriptive table of included studies.**

|  | *n*, (%) |
|---|---|
| **Collection method** |  |
| Electronic | 76 (89.41%) |
| Mixed | 5 (5.88%) |
| Pen and Paper | 4 (4.71%) |
| **Context** |  |
| Multicentre | 16 (18.82%) |
| Single Centre | 57 (67.06%) |
| Registry network | 3 (3.53%) |
| Not stated | 9 (10.59%) |
| **Country** |  |
| Netherlands | 29 (34.12%) |
| United States | 24 (28.24%) |
| Canada | 6 (7.06%) |
| Austria | 5 (5.88%) |
| UK | 4 (4.71%) |
| Denmark | 3 (3.53%) |
| Australia | 2 (2.35%) |
| Germany | 2 (2.35%) |
| Italy | 1 (1.18%) |
| Singapore | 1 (1.18%) |
| Spain | 1 (1.18%) |
| United States and Canada | 1 (1.18%) |
| Not stated | 7 (8.24%) |

Table 2 includes the full details of the electronic collection, storage and access of PROMs in the included studies.

## How are electronic PROMs collected?

Table 3 presents the different systems used for PROMs collection. The use of custom-built system/software to collect PROMs was common [15,23,24,26,32–34,39,44,45,47–51,53,55,56,59–63,65,67,69,70,72,74–78,81,86,87,93,95,96,98–106]. This included non-traditional methods of collecting data such as through television-based collection [24] or a gamified collection of PROMs (My-Pal app) [62]. Some studies used pre-existing systems, those that were not specifically built for PROMs collection such as Google online forms or RedCap were mentioned [28,36,52,57,58,64,71,80,88,89]. This included the incorporation of PROMs collection within EHRs such as using custom forms to collect PROMs [28,36]. Some studies did not specifically name the software or system used [29,31,36–38,40–43,66,68,73,84,85,90–92,94,97] or included non-electronic methods of collection [27,30,79,83].

## Where and how are PROMs data stored electronically?

Table 4 details the different electronic systems used for the storage of PROMs data. This includes external servers or databases, which can be custom-built or pre-existing systems [15,39–42,44–49,51–56,58–61,65,67,75–78,83,86,87,93,95,96,98–106] and those stored within EHR software platforms [50,57,68,73,79,81,85]. Some studies reported the use of both external servers or databases and within EHRs [84,88,97]. It was noted that storage of the data is not commonly reported within the publication of the routine use of PROMs [62–64,66,69–

**Table 2. Identified systems included in the scoping review exploring the electronic collection and capture of PROMs as part of standard treatment and care of children and young people in the hospital setting.**

| Name of system | Collection | Storage | Viewing |
|---|---|---|---|
| **Ambuflex** [15,51,78] | ePRO system but collection is done through eBoks which is a secure national email platform. | Stored in the portal and can be anonymized and transferred for further analysis | PRO overview is presented inside the electronic health record system for clinicians (no patient access or portal). Clinicians have direct access. Patient is contacted if necessary |
| **Electronic outcomes rating form** [65] | e-ORF, a paper form that uses a digital pen for electronic entry, build within the hospital's scheduling system | A digital pen is used to transfer the data to the hospital service and a PDF version of the e-ORF is produced via FusionForm software. This PDF data is stored in another server. This PDF data can then converted into CSV. | Scores appear directly in the patient's medical record. Clinicians have access. |
| **ePROtect** [59–61] | Web-based software | Stored within the portal | ePROtect- web-based patient portal can view scores with descriptions and a health care professional interface to review the data from all patients. Has automatic scoring of results. |
| **ePRO diary** [67] | An app | Within the software | Data extracted from the software by clinicians and patients |
| **KLIK** [75,76,86,87,95,96,98–105] | Web-based PROM portal | Stored in KLIK PROM portal | KLIK ePROfile retrieved from the website by clinicians. |
| **myHealthE (MHE) system** [63] | Web-based system | Not stated | Caregivers were presented with infographics based on their responses in the system |
| **PediQUEST system** [93] | Computer-based data collection system | In the PediQUEST system | Printed feedback reports and e-mail alerts (PQ reports and emails) to patients and clinicians |
| **QLIC-ON** [32,74,107] | Computerised questionnaires completed on laptops | Not stated but data can be exported to SPSS | QLIC-ON profile access on the web by clinicians. Can also be printed |
| **Television-based interactive patient care system** [24] | Patients/parents can use through their bedside television screens (in patients only) | University of Minnesota Research Clinical Data repository and linked to in patient records through electronic health records | Electronic health records by clinicians |
| **Tonic for Health Electronic Platform** [70,72] | Online webpage and app | Tonic cloud servers | Scores showed in either a laptop or paper printout to clinicians. |
| **Google Online form** [64] | Google online versions of questionnaires accessed through mobile devices | Not stated | Not stated |
| **LimeSurvey** [88] | Online, open source survey web application. Separate links sent to parent and patient. | Parent, patient and in-clinic questionnaire responses (and scored summary data where relevant) are uploaded to patients' electronic health records and in the Royal Children's Hospital Gender Service clinical database | Accessed via electronic health records |
| **Qualtrics** | | | |
| Robertson [71] | Large print paper version and electronic version | Not stated | Not stated |
| Tyack [89] | Qualtrics was used to administer ePROMs | Not stated | Not stated |
| **Quick response codes** [23] | Given at the end of their clinic visit and completed on mobile device with guidance from clinic staff when necessary | Not stated | Not stated |
| **REDCap** | | | |
| Leahy [52] | Web application | Downloaded from RedCap and processed with a excel macro for standardised reports to clinicians | Reports to clinicians using excel macros |
| Marker [57] | Web application | Data stored into EHRs and automatically pull scores from there. | Viewed in EHRs- automated scores and outcomes were documented in HER. Clinicians have real time direct access. |
| Mentrikoski [58] | Web application | RedCap database | Not stated |
| Smyth [80] | Web application | Not stated | Not stated |

*(Continued)*

**Table 2.** (Continued)

| Name of system | Collection | Storage | Viewing |
|---|---|---|---|
| Stratton [82] | Web application | Not stated | Not stated |
| **Life App** [35,69] | Web-based application through a computer-based health evaluation software (CHES) | Not stated | Clinicians have direct access |
| **MyPal-Child and MyPal-Carer app) [62]** | 2 mobile applications: the gamified MyPal-Child and MyPal-Carer App | Not stated | Reported data are visualised graphically for the medical staff via a web interface. |
| **KidsPRO [106]** | Electronic platform | within KidsPRO | KidsPRO application for children and families and within electronic health records for clinicians |
| **PROMIS-CAT [81]** | Electronic collection but not specified | EHRs | EHRs |
| **Kids-CAT [26,34]** | Electronic recording using a computer-adaptive test (CAT) | Not stated | Printed reports available to clinicians |
| **PROMIS [36]** | Part of EHRs | Not stated | Not stated |
| **Custom forms [28]** | In EHRs | In EHR using outpatient note templates to ensure that outcomes are measures in a discrete, mineable and reportable element. | EHRs |
| **myGeisinger [50]** | Web based patient portal | in EHRs (uses Epic) | Patient portal linked to EHRs |
| **Other** | | | |
| Bower [27] | Pen and paper | EHRs, transcribed by provider using smart forms in Epic | EHRs |
| Cheng [29] | Electronic collection but not specified | Not stated | Not stated |
| Cunningham [30] | Mixed (pen and paper and web-based data capture) | Directly linked to EHRs | EHRs with automated scoring and alerts |
| Eilander [31] | Not specified online survey | Not stated | Not stated |
| Anthony [25] | Not specified PROM portal | Not stated | Accessed by clinicians |
| Gerhardt [37] | Not specified electronic system | EHRs | Custom build PRO dashboard within EHRs (results available at individual and population level) that clinicians have direct access |
| Gmuca [38] | Not specified online survey | Not stated | Not stated |
| Gupta [40] | Not specified web-based platform | Scores are collected centrally in the Symptom management reporting database (SMRD) | Not stated |
| Gupta [41] | Not specified web-based platform | Scores are collected centrally in the Symptom management reporting database (SMRD) | Not stated |
| Hames [42] | Not specified web-based platform | Stored in the system | Not stated |
| Hanmer [43] | Not specified online survey | stored directly into EHRs (uses EPIC) | Viewed in EHRs, can be tracked over time and pulled into documentation to generate alerts to clinicians |
| Ng [66] | Not specified electronic survey and application (tele-health app) | Not stated | Automated scoring and transfer of data to the clinical team that incorporates differences between patient and parent scores |
| Perito [68] | Electronic collection but not specified | Data needed for registries required manual extraction and data transfer from electronic health records (EHR) into registry specific forms. | Not stated |
| Ross [73] | Not specified web-based platform | EHRs | EHRs |
| Sheikh[79] | Pen and paper | Results were uploaded onto the electronic medical records on the day it was completed. | EHRs and clinicians have direct access |
| Swales [83] | Paper collection then manually entered on the website | In the system for multi-site analysis | Not stated |

(*Continued*)

**Table 2.** (Continued)

| Name of system | Collection | Storage | Viewing |
|---|---|---|---|
| Taxter [84] | Not specified web-based platform | Build a clinic note templates into electronic health records to capture Childhood Arthritis and Rheumatology Research Alliance registry data | Electronic health records for clinician access, flowsheet function also means it can be shared with patients. |
| Taxter [85] | Electronic collection but not specified | electronic health records | Not stated |
| Valles [90] | Electronic collection but not specified | Not stated | Not stated |
| Van Sonsbeek [97] | Not specified online survey | Routine Outcome Monitoring (ROM) system or EHRs | Results can be directly access from the ROM system or in the electronic health records (a day after) |
| Wang [94] | Not specified web-based platform | Not stated | Feedback system giving real-time delivery of BOQ+P results to clinicians prior to a patient's encounter |
| Yao [92] | Not specified online survey | Not stated | Not stated |

72,74,80,82,89–92,94]. There were some explicit mentions on the structure of the data or whether it could be used for further analyses and at different population breakdowns. For example, one study mentioned the data were stored in "discrete, mineable and reportable elements" [28] and other studies mentioned an "anonymised version" of the PROM results was available through the Ambuflex portal [15,51,78].

## Where and how are PROMs results viewed electronically and how does this influence access?

Table 5 provides information about the different systems used for viewing PROMs results. PROMs results were accessed through external software/systems which have been custom-

**Table 3. Table of identified electronic platforms by categories.** Each named system is only counted once even if there are multiple publications.

| Custom-built systems/software (n = 17) | Pre-existing systems/software and within EHRs (n = 6) | Other not specified/mixed collection/pen and paper (n = 22) |
|---|---|---|
| Ambuflex [15,51,78] | PROMIS- part of electronic health records (EHRs) [36] | Other [29–31,36–38,40–43,66,68,73,79,83–85,90–92,94,97] |
| Electronic outcomes rating form [65] | Custom forms [28] | |
| ePROtect [59–61] | Google Online form [64] | |
| ePRO diary [67] | LimeSurvey [88] | |
| KLIK [75,76,86,87,95,96,98–105] | Qualtrics [71,89] | |
| myHealthE [63] | RedCap [52,57,58,80,82] | |
| PediQUEST system [93] | | |
| QLIC-ON [32,74,107] | | |
| Television-based interactive patient care system [24] | | |
| Tonic for Health Electronic Platform [70,72] | | |
| Quick response codes [23] | | |
| Life App [35,69] | | |
| MyPal-Child and MyPal-Carer app (gamified collection) [62] | | |
| KidsPRO [106] | | |
| PROMIS-CAT [81] | | |
| Kids-CAT [26,34] | | |
| myGeisinger patient portal [50] | | |

**Table 4. Electronic systems for storage of PROMs data.** Each named system is only counted once even if there are multiple publications.

| External server/database (*n* = 14) | Within EHRs (*n* = 12) | Both external server and within EHRs (*n* = 3) | Not stated (*n* = 18) |
|---|---|---|---|
| Ambuflex [15,51,78] | Custom forms [28] | LimeSurvey [88] | myHealthE [62] |
| Electronic outcomes rating form [65] | PROMIS-CAT [81] | Other [84,97] | QLIC-ON [73] |
| ePROtect [59–61] | myGeisinger [50] | | Google Online form [64] |
| ePRO diary [67] | RedCap [57] | | Qualtrics [71,89] |
| KLIK [75,76,86,87,95,96,98–105] | Other [27,30,37,43,68,73,79,85] | | Quick response codes [23] |
| PediQUEST system [93] | | | Life App [69] |
| Television-based interactive patient care system [24] | | | MyPal-Child and MyPal-Carer app (gamified collection) [62] |
| KidsPRO [106] | | | Kids-CAT [34] |
| RedCap [52,58] | | | PROMIS- part of electronic health records (EHRs) [36] |
| Tonic Cloud Servers (Tonic for Health Electronic Platform) [70,72] | | | RedCap [80,82] |
| Other [40–42] | | | Other [25,29,31,38,66,90,92,94] |

built for PROMs implementation such as KLIK or Ambuflex, or systems that are widely available such as emails or Microsoft Excel (using macros) [32,33,59–63,66,67,70,72,74–77,86,87, 91,93,95,96,98–105]. Results could also be accessed within EHRs [15,37,43,50,51,57,65,73,78, 79,81,84,88] or through both external software/systems and within EHRs [26,34,97,106].

Results were commonly viewed at an individual level and presented in a graphical format irrespective of system. Clinicians typically had direct access to the individual patients' PROM results and could share findings with patients/parents during consultations. However, we also found some studies reporting that clinicians needed to follow additional steps to gain access to the results, such as one study outlining that results needed to be printed to be visible to clinicians even with electronic collection and/or storage [26,34,93]. In others, the systems allowed patients/families to have direct access to their results using their own portals [50,67,84,106]. Several also reported automatic scoring of PROMs [50,57,59–61] with triggers to be sent directly to patients and/or clinicians for further action or advice via the portals [59–61].

**Table 5. Type of system for viewing electronic PROMs.** Each named system is only counted once even if there are multiple publications.

| External software/system (*n* = 9) | Within EHRs (*n* = 15) | Both external software/system and within EHRs (*n* = 2) | Not stated (*n* = 21) |
|---|---|---|---|
| ePROtect [59–61] | Ambuflex [15,51,78] | KidsPRO [104,106] | Kids-CAT (printed) [26,34] |
| ePRO diary [67] | Electronic outcomes rating form [65] | Other [97] | PediQUEST system (printed) [93] |
| KLIK [75,76,86,87,95,96,98–105] | Television-based interactive patient care system [24] | | Google Online form [64] |
| myHealthE [63] | LimeSurvey [88] | | Qualtrics [71,89] |
| QLIC-ON [32,33,74] | PROMIS-CAT [81] | | Quick response codes [23] |
| Tonic for Health Electronic Platform (also printed) [70,72] | Custom forms [28] | | Life App [35,69] |
| RedCap [52] | RedCap [57] | | PROMIS- part of electronic health records (EHRs) [36] |
| MyPal-Child and MyPal-Carer app [62] | myGeisinger [50] | | RedCap [58,80,82] |
| Other [66] | Other [27,30,37,43,73,79,84] | | Other [25,29,31,38,40–42,68,83,85,90,92,94] |

How connected are electronic collection, storage and access?

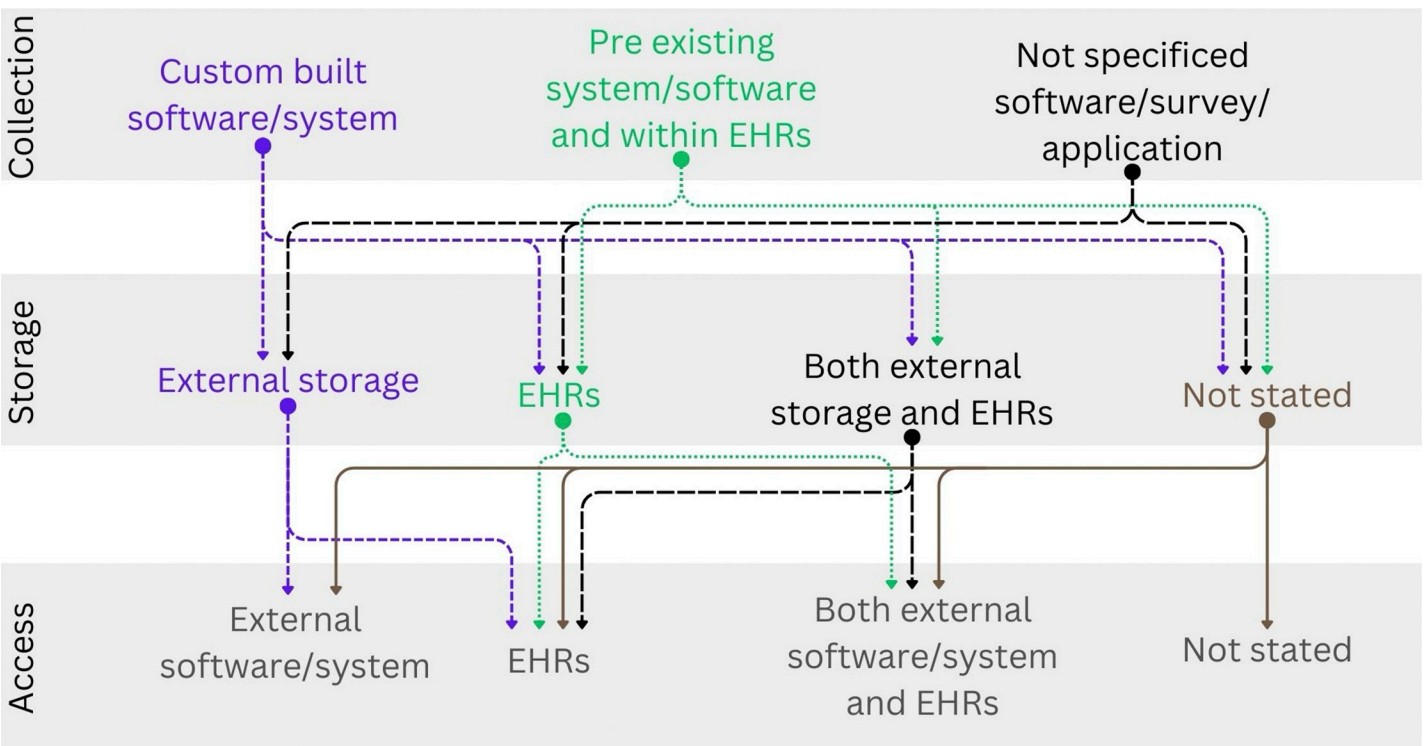

Note: EHRs - Electronic Health Records

**Fig 2. Diagram of how the different categories of collection, storage and access map on to one another.**

Fig 2 shows how the different methods of collection, storage and access all relate to one another. For some studies, the collection, storage and viewing used the same system such as KLIK PROM portal [75–77,86,87,91,95,96,98–105] or ePROtect [59–61] which typically included an external custom-built portal for collection and viewing and their own external storage database for the storage of the PROMs data. However, there was great variability in the configuration of these electronic systems in relation to the collection, storage and viewing of the PROMs data. For example, Ambuflex [15,51,78] and myGeisenger [50] collected PROMs through their portal but data were stored within their server and could be viewed within EHRs, whereas KidsPRO [106] collected PROMs through their portal, data were stored within their portal but could be viewed within their portal and within EHRs.

Our review also identified studies that used pen and paper methods or mixed methods of collecting PROMs but stored the results in electronic formats, for example, one study reported staff manually transcribing the paper responses into EHRs using smart forms [27] and another study used a digital pen to translate the paper responses into electronic format so they could be stored in an external server that was linked to EHRs [65]. Some studies also reported that even though PROMs were collected and stored digitally, the results could only be accessed through printed formats[27,34] with some also available via emails [93].

## Discussion

### Key findings

This review demonstrates that there is no standardised or uniform method for the implementation of electronic PROMs in routine care of CYP in the hospital setting. There is a range of

electronic systems and different types of set-up for the collection, storage and access of electronic PROMs in routine care. Of note, the electronic storage of PROMs can influence how the results are accessed and viewed by clinicians and patients/carers, and how the data are stored. The variability in the electronic capture of PROMs is likely to affect ease of access to the results and ability to apply the data in clinical care but also the potential to use deidentified PROMs data for secondary purposes. Consequently, the variability in the electronic collection and capture of PROMs will have a downstream consequence on the potential value of PROMs at patient, immediate clinical level and for population level analytics.

## Opportunities and challenges of electronic PROMs

The use of electronic platforms for collecting and the integrating PROMs into EHRs can help facilitate the implementation of PROMs in routine care [7,13]. However, the exclusive use of digital systems for collection may hinder patients or carers who may not be able to access, or be comfortable using this type of technology [108]. It is important that the use of electronic platforms in the implementation of PROMs in routine care is beneficial across all patient groups and does not introduce unintended inequalities in care. The findings of this review suggests that electronic storage such as the integration into EHRs can act as a bridge for traditional (e.g. pen and paper) or even mixed methods of collection and can broaden the flexibility and reach of collecting PROMs in routine care, this might help address issues of digital literacy and restricted access to technology, maximising the chances that PROMs are beneficial for all patient groups.

Collecting PROMs in routine care of CYP and integrating the data into EHRs expands the available routinely collected data beyond clinical information to include the patient voice. Routinely collected data such as EHR data typically only include data generated by clinicians or administrators so they are likely to focus on the clinical or process outcomes of care [109] whereas the use of PROMs and other patient reported data can bring the focus back to the patient, facilitating person-centred care [110]. Furthermore, similar to other routinely collected data, EHR data including PROMs can be used for secondary purposes such as research, clinical registries and service improvement initiatives [3,4,10–12] which is crucial to achieving a learning healthcare system. Yet these types of data can sometimes be difficult and time consuming to process, may be incomplete or inaccurate or may be difficult to share across different systems [111–113]. The ability to easily process and share these types of data across different systems is also known as interoperability [114,115]. Interoperability within IT health systems such as EHRs can be achieved at different levels depending on the type of data and the systems being used [114]. The ability to exchange information between different IT systems is known as technical interoperability and can be achieved through transport standards such as HL7 FHIR [114,115]. However, communication between systems is not essential for data sharing; semantic interoperability allows for different IT systems to understand and comprehend one another by collecting data in the same structure using common data standards such as SNOMED codes for clinical coding of diagnoses [114,115]. The review identified a substantial number of vendors and systems supporting the implementation of PROMs in routine care, fostering siloed approaches to implementation. The use of these independent vendors or systems may run the risk of healthcare providers falling into 'vendor lock-in' or difficulties in sharing the data across the different systems. Although exploring the ideal platform is beyond the scope of this review and factors such as user interface, costs, integration within existing processes should also be taken into account, this review suggests that technical aspects of the platform should also be considered. This includes exploring how platforms may use transport standards such as HL7 FHIR, or collect PROMs data using common data standards. The latter

aims to ensure that the data collected and captured are accessible, shareable and usable outside the individual platforms. These interoperable solutions can help facilitate the ability to maximise the value of PROMs in routine care. Yet it is currently unclear whether these technical aspects of interoperability are considered in the implementation of PROMs or when selecting these systems and vendors for implementation.

A previous framework suggests interoperable IT solutions are needed to facilitate a system-wide adoption of PROMs implementation at national and international levels [12]. The findings of this review support this recommendation and suggest that such considerations should be encouraged even at early stages of the implementation process. The collection and use of PROMs as part of standard care of children and young people is a fast-evolving field and without factoring technical considerations to enable interoperability, both technical and semantic, the implementation of PROMs in routine care may lead to divided approaches even within the same hospital. This may hinder the potential for scalability of PROMs and limit the potential value of PROMs within that same hospital but also across different health care settings. Consequently, the consideration of interoperable solutions to the data collection specially at early stages can help ensure that the data collected is easy to manage and is usable reducing the risk of overabundance of unusable data and facilitating the scalability of PROMs in routine care. As such, further research is needed to explore the technical aspects required to leverage these electronic systems such as identifying common data standards to ensure that the PROMs data collected are interoperable (shareable), accessible, usable at an individual, organisational, national and international level. In an environment where healthcare systems data such as diagnostic codes are more advanced and are increasingly employed for data-driven approaches such as artificial intelligence, this will better align with PROMs data being potentially able to contribute. Although beyond the scope of this manuscript, this also presents new challenges such as privacy concerns in relation to PROMs, particularly if the data are unstructured and contain e.g. free text responses. Further research is needed on the best policies for privacy and security when employing these tools which should also be considered when choosing a platform for PROMs collection and capture.

## Strengths/Limitations

To the best of our knowledge this is the first review to document the electronic platforms that are in use to implement PROMs in the routine hospital care of CYP. Furthermore, it included studies that are further in the implementation stage and are routinely using PROMs as part of standard treatment and care of CYP.

However, the technical aspects of the implementation of PROMs in routine care including specification of the electronic collection and capture may be further detailed in a different publication which was out of the scope of the original review protocol. Additionally, anecdotal evidence suggests that it is unlikely that all technical details of these processes are published in academic literature. Consequently, it is possible that this review is underrepresenting the technical aspects and considerations in the implementation of PROMs in routine care of CYP using electronic platforms for collection and capture. Further research is needed for a systematic capture of the literature which focuses on the technical considerations when using electronic platforms for implementation of PROMs in routine care.

## Conclusion

Electronic platforms for the collection and capture of PROMs in routine care of CYP are popular but there is great variety in the platforms used and many different configurations of how this has been implemented. The digital and interoperable storage of PROMs is particularly

significant as it has potential to broaden the reach, facilitate access and catalyse additional approved uses of the data for service improvement, research and health policy. However, further research to explore the technical considerations needed for implementation of these electronic platforms, including creating interoperable solutions, is needed to maximise the potential value and support the scalability of PROMs in routine care.

## Supporting information

**S1 File. Full inclusion and exclusion criteria.**
(DOCX)

**S2 File. Example of search strategy (Medline).**
(DOCX)

**S3 File. Extraction form.**
(DOCX)

**S4 File. Full description of included studies.**
(DOCX)

## Acknowledgments

All research at Great Ormond Street Hospital NHS Foundation Trust and UCL GOS Institute of Child Health is supported by the National Institute for Health Research Great Ormond Street Hospital Biomedical Research Centre (NIHR GOSH BRC). The views expressed are those of the authors and not necessarily those of the NHS, the NIHR or the Department of Health.

We would like to thank Katherine Terrell, Paula Kelly, Heather Chesters, Faith Gibson, Geralyn Oldham and Debbie Sell for their work on the wider PROMs/PREMs scoping review. This work would not have been possible without their contributions.

## Author Contributions

**Conceptualization:** Anne Alarilla, Neil J. Sebire, Josh Keith, Mario Cortina-Borja, Jo Wray, Gwyneth Davies.

**Data curation:** Anne Alarilla, Jo Wray, Gwyneth Davies.

**Formal analysis:** Anne Alarilla, Jo Wray, Gwyneth Davies.

**Project administration:** Anne Alarilla.

**Supervision:** Jo Wray, Gwyneth Davies.

**Validation:** Anne Alarilla, Jo Wray, Gwyneth Davies.

**Visualization:** Anne Alarilla.

**Writing – original draft:** Anne Alarilla.

**Writing – review & editing:** Anne Alarilla, Neil J. Sebire, Josh Keith, Mario Cortina-Borja, Jo Wray, Gwyneth Davies.

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
