## [Decision Letter · Decision Letter 0]

1 Oct 2024

PDIG-D-24-00351

A scoping review of the electronic collection and capture of patient reported outcome measures for children and young people in the hospital setting

PLOS Digital Health

Dear Dr. Alarilla,

Thank you for submitting your manuscript to PLOS Digital Health. After careful consideration, we feel that it has merit but does not fully meet PLOS Digital Health's publication criteria as it currently stands. Therefore, we invite you to submit a revised version of the manuscript that addresses the points raised during the review process.

Please submit your revised manuscript within 30 days Oct 31 2024 11:59PM. If you will need more time than this to complete your revisions, please reply to this message or contact the journal office at digitalhealth@plos.org. Please include the following items when submitting your revised manuscript:

We look forward to receiving your revised manuscript.

Kind regards,

Jennifer N Avari Silva, MD

Section Editor

PLOS Digital Health

Jennifer Silva

Section Editor

PLOS Digital Health

Journal Requirements:

Additional Editor Comments (if provided):

Reviewers' comments:

Reviewer's Responses to Questions

**Comments to the Author**

1. Does this manuscript meet PLOS Digital Health’s publication criteria? Is the manuscript technically sound, and do the data support the conclusions? The manuscript must describe methodologically and ethically rigorous research with conclusions that are appropriately drawn based on the data presented.

Reviewer #1: Yes

2. Has the statistical analysis been performed appropriately and rigorously?

Reviewer #1: N/A

3. Have the authors made all data underlying the findings in their manuscript fully available (please refer to the Data Availability Statement at the start of the manuscript PDF file)?

Reviewer #1: Yes

4. Is the manuscript presented in an intelligible fashion and written in standard English?

Reviewer #1: Yes

5. Review Comments to the Author

Reviewer #1: I congratulate Alarilla et al. on their submission, A scoping review of the electronic collection and capture of patient reported outcome measures for children and young people in the hospital setting. 

This is a very comprehensive review of the current state patient report outcome measure assessments as recorded or stored in the electronic health record. 

The authors conducted a review twice, of 7 databases, first in February 2021 and again in April 2023, for studies that used electronic systems as part of patient reported outcome measure assessments in secondary and tertiary care settings. They focused on studies where data was collected from patients aged birth to 25 years, patient proxies, or clinicians working with the aforementioned patient population. 

They identified 85 studies out of 372 that met their inclusion criteria. 

The authors stated that the aim of their scoping review was to map the available evidence of the electronic collection and capture of PROMs used as part of standard care of children and young people (CYP) focusing on technical aspects. 

• What are the electronic systems used for PROMs collection? 

• What are the electronic systems used for the storage of PROMs data? 

• How are PROMs data accessed electronically? 

• How do the electronic collection, storage and access relate to one another?

The authors tables and accompanying text answer the above questions in detail, identifying uses, challenges and benefits of PROMs. 

The authors found that though popular, there is significant variability in all aspects of use of PROM, from data collection, to storage, implementation and access. 

They suggest, “further research to explore the technical considerations needed for implementation of electronic platforms, including creating interoperable solutions,” so the value of these assessments can be “maximized…and support the scalability of PROMs in routine care.”

Minor changes: 

Abstract: The abstract appears to be incomplete. In the introduction, the last line appears to be cut off. “Yet little is known about…”

Last line, page 21 of manuscript, there is a period and comma. 

Comments/Questions: 

1/ If you were to make suggestions about creating an ideal platform, what would you suggest? How would you address the challenges of universal digital literacy or access to such technology? When hybrid forms are used, there is such as burden on staff (transcribing paper responses, etc), and challenges to use the data, as identified in your manuscript. What would be an ideal platform in CYP to allow easy access, storage, use and implementation of this data?

One current conversation in digital health is how does one manage the overabundance of data? Do you envision this happening?

Does AI and other machine learning platforms have a space in PROMs, while ensuring patient privacy?

6. PLOS authors have the option to publish the peer review history of their article (what does this mean?). If published, this will include your full peer review and any attached files.

**Do you want your identity to be public for this peer review?** For information about this choice, including consent withdrawal, please see our Privacy Policy.

Reviewer #1: No

---

## [Decision Letter · Decision Letter 1]

19 Nov 2024

A scoping review of the electronic collection and capture of patient reported outcome measures for children and young people in the hospital setting

PDIG-D-24-00351R1

Dear Ms Alarilla,

We are pleased to inform you that your manuscript 'A scoping review of the electronic collection and capture of patient reported outcome measures for children and young people in the hospital setting' has been provisionally accepted for publication in PLOS Digital Health.

Best regards,

Jennifer N Avari Silva, MD

Section Editor

PLOS Digital Health

**Additional Editor Comments (if provided):**

**Reviewer Comments (if any, and for reference):**

Reviewer's Responses to Questions

**Comments to the Author**

1. If the authors have adequately addressed your comments raised in a previous round of review and you feel that this manuscript is now acceptable for publication, you may indicate that here to bypass the “Comments to the Author” section, enter your conflict of interest statement in the “Confidential to Editor” section, and submit your "Accept" recommendation.

Reviewer #1: All comments have been addressed

2. Does this manuscript meet PLOS Digital Health’s publication criteria? Is the manuscript technically sound, and do the data support the conclusions? The manuscript must describe methodologically and ethically rigorous research with conclusions that are appropriately drawn based on the data presented.

Reviewer #1: Yes

3. Has the statistical analysis been performed appropriately and rigorously?

Reviewer #1: N/A

4. Have the authors made all data underlying the findings in their manuscript fully available (please refer to the Data Availability Statement at the start of the manuscript PDF file)?

Reviewer #1: Yes

5. Is the manuscript presented in an intelligible fashion and written in standard English?

Reviewer #1: Yes

6. Review Comments to the Author

Reviewer #1: No additional comments

7. PLOS authors have the option to publish the peer review history of their article (what does this mean?). If published, this will include your full peer review and any attached files.

**Do you want your identity to be public for this peer review?** For information about this choice, including consent withdrawal, please see our Privacy Policy.

Reviewer #1: No
